# Retrieval of Hyperspectral Information from Multispectral Data for Perennial Ryegrass Biomass Estimation

**DOI:** 10.3390/s20247192

**Published:** 2020-12-15

**Authors:** Gustavo Togeiro de Alckmin, Lammert Kooistra, Richard Rawnsley, Sytze de Bruin, Arko Lucieer

**Affiliations:** 1School of Technology, Environments & Design, University of Tasmania-Geography and Spatial Sciences, Hobart, TAS 7001, Australia; gustavo.alckmin@wur.nl (G.T.d.A.); Arko.Lucieer@utas.edu.au (A.L.); 2Laboratory of Geo-Information Science and Remote Sensing, Wageningen University & Research, P.O. Box 47, 6700 AA Wageningen, The Netherlands; sytze.debruin@wur.nl; 3Centre for Dairy, Grains and Grazing, Tasmanian Institute of Agriculture, Cradle Coast Campus, 16–20 Mooreville Road, Burnie, TAS 7320, Australia; Richard.Rawnsley@fonterra.com

**Keywords:** vegetation indices, spectral resampling, continuum-removal, parametric-regression, spectral simulation, machine learning, random-forest

## Abstract

The use of spectral data is seen as a fast and non-destructive method capable of monitoring pasture biomass. Although there is great potential in this technique, both end users and sensor manufacturers are uncertain about the necessary sensor specifications and achievable accuracies in an operational scenario. This study presents a straightforward parametric method able to accurately retrieve the hyperspectral signature of perennial ryegrass (*Lolium perenne*) canopies from multispectral data collected within a two-year period in Australia and the Netherlands. The retrieved hyperspectral data were employed to generate optimal indices and continuum-removed spectral features available in the scientific literature. For performance comparison, both these simulated features and a set of currently employed vegetation indices, derived from the original band values, were used as inputs in a random forest algorithm and accuracies of both methods were compared. Our results have shown that both sets of features present similar accuracies (root mean square error (RMSE) ≈490 and 620 kg DM/ha) when assessed in cross-validation and spatial cross-validation, respectively. These results suggest that for pasture biomass retrieval solely from top-of-canopy reflectance (ranging from 550 to 790 nm), better performing methods do not rely on the use of hyperspectral or, yet, in a larger number of bands than those already available in current sensors.

## 1. Introduction

Precision agriculture (PA) and Remote Sensing (RS) employ spectral observations to estimate biophysical and biochemical properties of vegetation canopies [1]. With the recent popularization of Unmanned Aerial Systems (UASs) for PA, farmers can now employ Multispectral (MS) cameras for spectral data acquisition, allowing for a high degree of flexibility in terms of data collection interval and custom-designed sensing systems [2]. Such systems can potentially be employed in precision pasture management as a monitoring tool [3], mapping key attributes such as biomass [4]. This potential, however, is dependent on the optimal design of sensors [5], particularly on the number, position, and width of spectral bands as well as its radiometric accuracy [6], aiming to provide high-quality predictors.

Currently, while constrained to MS data, PA practitioners rely on Vegetation Indices (VIs), rather than individual band reflectances, as enhanced predictors of the canopy biophysical and biochemical properties [7,8]. Overall, indices are summary measurements of broader and more complex systems, acting as indicators of an underlying phenomena. An index, in essence, summarizes a fraction of the spectral variability while tracking changes in the canopy [9] without necessarily resorting to the complete spectral information. In this sense, VIs are a valuable trade-off between a small number of bands and the full information available in a spectral signature [10].

Many indices, however, are prone to become irresponsive (i.e., to saturate) at high levels of biomass (e.g., Leaf Area Index (LAI) ≥4) [11,12], regardless of a continuous response in different parts of the spectrum [13]. Furthermore, a large number of indices are available and proposed for different applications [14]. Consequently, this approach requires domain knowledge for optimal vi selection [15] and measurements at specific bands, requiring purposely built MS sensors, which are usually either not commercially available or cost-prohibitive for commercial applications.

In a complementary way, Hyperspectral (HS) data have been shown to provide a higher level of accuracy when estimating different canopy properties [16,17], particularly those which are functionally linked to narrowband absorption features, provided that the sensor spectral resolution is smaller than the observed spectral feature [18]. As a drawback, the necessary sensor to collect such data is usually more complex and costly [3]. Furthermore, broadband absorption features may overlap and mask the discrete and informative narrowband spectral signal [19], impairing straightforward analysis and the added benefits of HS data [20]. Last, hyperspectral measurements display a high level of multicollinearity [21], providing redundant information, particularly in contiguous bands [22]. Yet, while the individuals band reflectance per se are highly redundant, other spectral features such as rates of changes (i.e., derivatives), areas (i.e., integrals), and Continuum-Removal (CR) transformations may best capture, disentangle, and enhance the predictive ability of hyperspectral data [23,24].

The constraints of VIs and HS are often presented as a dichotomy: while the first is seen as coarse and of limited efficiency [25], the latter is presented as highly costly and complex yet able to map subtle changes in biochemical and biophysical properties [20]. In many instances, VIs provide a partial understanding of the full spectral behavior, only directly displaying a fraction of the information contained in a spectral signature.

Within the spectral range of most commercial sensors (i.e. visible to near-infrared, 500–900 nm), the spectral signature of canopies is largely driven by broadband features (e.g., chlorophyll content and LAI) [26], corresponding to known response-functions [27]. It is, thus, reasonable to hypothesize that through a few sparse spectral measurements, the in-between bands can be interpolated and the underlying hyperspectral information can be retrieved, provided that adequate functions are chosen to describe the overall spectral behavior.

This *proposed* method would allow the reconstruction the spectral signature while taking advantage of the sparse nature of bands in commercially available MS cameras, and the generation of non-directly observed spectral bands and features. A similar approach was employed by Jongschaap and Booij [28], who simulated the wavelength (λ) range from 660–810 nm in order to estimate the red-edge position, using a form of logistic function. More complex approaches, such as Bayesian [29] and multivariate methods [30], have also been successfully employed to retrieve hyperspectral information from multispectral measurements.

Specifically for pasture biomass estimation, Mutanga and Skidmore [16] described the use of *CR* to derive vegetation indices while consequently reporting better performances than the usual normalized ratio indices. Furthermore, using an exhaustive search procedure, several authors have proposed a red-edge-normalized ratio index to estimate LAI and biomass [31,32,33]. These indices, however, cannot be directly extracted from commercially available MS sensors (either satellite or UAS-based), as the necessary bands are not measured. Yet, although these indices are not directly available from MS observations, accurately simulated spectra would allow their indirect estimation, and possibly lead to better estimations of biomass.

In summary, this study aims to (i) simulate hyperspectral data (550–790 nm) from multispectral measurements which can be recovered from a commercially available camera; (ii) from the the simulated spectral signature, derive a set of synthetic high-performance indices and high-performing spectral features found in the scientific literature [16]; and (iii) assess predictions of perennial ryegrass (*Lolium perenne*) biomass of the *proposed method* against a common subset of commercially employed vegetation indices, referred to as the *current method*.

The performance of these two methods will be further contrasted based on a cross-validation and spatial cross-validation procedure [34], granting insight on their performance in known and new locations. As a final analysis, for a deeper understanding of the performance analysis, indices and spectral features will be clustered through a dendrogram, providing an in-depth analysis of their redundancy level.

If successful, the *proposed method* would expand the number of VIs and spectral features generated from readily available MS cameras, matching more costly and complex HS sensors when applied to top-of-canopy reflectance measurements for pasture management. Finally, the accurate retrieval of HS information of vegetated canopies, purely based in MS data, presents an important argument to the optimal number of spectral bands for sensor design and whether hyperspectral data, and its derived features add value to the task of pasture biomass retrieval.

## 2. Methods

### 2.1. Experimental Layout

The experimental field design (concerning nitrogen levels and regrowth periods) was chosen to generate a wide range of biophysical and biochemicals properties while being representative of typical pasture management conditions of each location [35,36]. The trial was performed at the Tasmanian Dairy Research Facility (Elliot (41${}^{{}^{\circ}}$04${}^{\prime }$57.7${}^{\prime \prime }$ S, 145${}^{{}^{\circ}}$46${}^{\prime }$22.0${}^{\prime \prime }$ E)) (Australia) and at three research stations (Goutum, Vredepeel and Zegveld) (Goutum 53${}^{{}^{\circ}}$10${}^{\prime }$38.8${}^{\prime \prime }$ N, 5${}^{{}^{\circ}}$46${}^{\prime }$27.0${}^{\prime \prime }$ E), Vredepeel 51${}^{{}^{\circ}}$32${}^{\prime }$57.9${}^{\prime \prime }$ N, 5${}^{{}^{\circ}}$51${}^{\prime }$54.0${}^{\prime \prime }$ E), and Zegveld 52${}^{{}^{\circ}}$08${}^{\prime }$32.9${}^{\prime \prime }$ N, 4${}^{{}^{\circ}}$50${}^{\prime }$23.5${}^{\prime \prime }$ E) affiliated to Wageningen University & Research (The Netherlands).

In Australia, the experimental layout was an array of 30 rainfed perennial ryegrass plots (dimensions of 2.0 × 7.5 m, with a 0.35 m border at each side of the plot’s longitudinal axis), arranged as two rows by 15 columns. Within each plot, six spectra–biomass sample pairs were collected. Plots were managed under five different *Nitrogen (N)* levels (0, 25, 50, 75, and 100 kg N/ha) and two regrowth intervals of approximately 15 and 30 days, with three pseudo-replicates of these combinations. Nitrogen levels were chosen based on previous research [36].

Data collection campaigns took place on 18 December 2016; 6 February, 29 April, 22 October, and 28 November 2017; and 11, 17, and 24 November 2018. In total, 1200 spectra–biomass sample pairs were collected in Australia (Figure 1).

In the Netherlands, each experimental site contained 24 plots (dimensions of 20 × 20 m) and spectra–biomass sample pairs were taken at uniform canopy cover and in the central area of each plot. The layout was a factorial combination of three N fertilization levels (0, 180, and 360 kg N/ha per year), four mowing intervals (in a cycle of four weeks, when then all plots were mown), and two pseudo-replicates of this combination. Each experimental setup was located within a different soil type (either clay, sandy, or peat). Data collection spanned from May to October 2018. From the second half of July until September, plant growth was constrained due to a prolonged heatwave and drought. In total, 216 spectra–biomass sample pairs were collected in the Netherlands (Figure 1).

### 2.2. Spectral Data

An identical data collection protocol was performed across sites and dates (Figure 2a–c. Spectral measurements were taken during clear-sky periods at around solar noon ±2 h, lasting for around 30 min in the Netherlands to one and half hours in Australia. To avoid a systematic illumination effect across plots, the order of spectral measurements of plots was randomized. An ASD FieldSpec^®^ 3 was employed in the Netherlands, and a FieldSpec^®^ Handheld 2 and a FieldSpec^®^ 4 (Malvern PanAnalytical-Colorado, Boulder, CO, USA) were employed in Australia. The instruments had no attached foreoptics (i.e., bare fiber; field of view: 25°) and were configured using the following setup (i.e., number of scans): 60 for white reference (Spectralon™. Labsphere-New Hampshire, North Sutton, NH, USA), 60 for dark-current, 30 per measurement, and five measurements of each target. Measurements were taken at nadir, from approximately one meter height, resulting in a circular footprint of approximately 0.44 m diameter (Figure 2a). The instrument was recalibrated whenever the white reference measurement deviated from a straight line centered at one or a maximum time limit of seven minutes between recalibrations was reached, whichever occurred first.

### 2.3. Reference Observations

The sensor’s footprint was mechanically defoliated to a specific residual height (i.e., 50 mm) and stored in perforated plastic oven bags (Figure 2a,b). Such residual height corresponds to the best practice for perennial ryegrass management, reflecting a high level of light interception and common residual grazing height, consequently best portraying an operational scenario. The harvested material was immediately refrigerated and transported from the experimental sites to a forced-air oven, where it was dried for 48 h at 65 °C and weighed (Figure 2c). In total, 1416 spectra–biomass sample pairs were collected.

### 2.4. Data Analysis

Data was analyzed in R (version 4.0.2) [37], and for reproducibility purposes data analysis R operations are introduced by the corresponding package::function format (typewriter typeface and accompanied by the double colon operator, i.e., the scope resolution operator).

### 2.5. Reference Analysis

*Comparability between locations*: biomass measurements were tested for a series of post hoc tests (i.e., analysis of variance) to ensure that data collected between locations was comparable, testing whether average values (μ) and variances ($\sigma {}^{2}$) were significantly different (H_0_: $\mu {}_{i}$ = $\mu {}_{j}$ at α = 0.05). Initially, the Bartlett Test of Homogeneity of Variances (stats::bartlett.test) was performed, followed by Kruskal–Wallis (stats::kruskal.test) and Dunn’s Multiple Comparisons (FSA::dunnTest), following the procedures described in Dinno [38].

A Principal Component Analysis (PCA) was employed (FactoMineR::PCA) to decompose the variability of the MS data and visualize (factoextra::fviz_pca_ind) whether different locations grouped together in clusters, providing evidence whether or not spectral observations were comparable (i.e. not highly influenced by the soil background). For visualization purposes, ellipsis were drawn covering 95% of the observations of each location.

#### 2.5.1. Hyperspectral Simulation

Spectral data were convolved (hsdar::spectralResampling) to the same specifications as a commercial multispectral sensor (Parrot Sequoia, Ile-de-France, France). Spectral resolution and the spectral response were extracted from its technical sheet using WebPlotDigitizer [39]. Four bands were generated: B_green_ (λ = 550 ± 40 nm), B_red_ (λ = 660 ± 40 nm), B_red edge_ (λ = 735 ± 10 nm), B_NIR_ (λ = 790 ± 40 nm).

*Piecewise equation*: The proposed method for hyperspectral simulation incorporated aspects of two previous studies [28,30]. The first approach was presented in Zeng et al. [30], where the authors use piecewise polynomials (i.e., splines) to interpolate the in-between spectra of a MS camera. Complementary, Jongschaap and Booij [28] fitted a sigmoid function to the reflectance measurements of a CropScan™MSR87 (CropScan-Minnesota, Rochester, NY, USA), showcasing the necessity of different functions types for different ranges of the spectra.
(1)$$f\left(\lambda \right)=\left\{\begin{array}{c} \mathrm{linear}\;\mathrm{function}\;\lambda <685\\ \mathrm{logistic}\;\mathrm{function}\;\lambda >685 \end{array}\right.$$

In the *proposed method*, hyperspectral simulation was based on an interpolation approach using a piecewise assembly of functions (Equation (Equation 1)) to the convolved multispectral dataset (Figure 3). In its first interval (i.e., sub-domain), the sub-function employed was a linear equation (Equation (Equation 2)), in which the slope (i.e., β) was calculated using the B_green_ and B_red_ bands (550 and 660 nm, respectively). The second interval was based on a three parameter logistic function (i.e., K, $R_{680}$ or B_red_, and C-Equation (Equation 3)). Two of the parameters (i.e., K and B_red_) were the original band values, corresponding to the starting value (i.e., B_red_) and asymptote (i.e., B_NIR_) of the logistic function, respectively. The remaining and unknown parameter C (steepness rate) was found through a Nelder–Mead optimization approach (stats::optim) using three bands (B_red_, B_red edge_ and B_NIR_). For each spectral observation, a set of parameters was computed and used to generate the simulated hyperspectral reflectance values (550–790 nm). No constraints were applied to ensure that the function was either continuous or differentiable. The piecewise function breakpoint was set to 685 nm (Equation (Equation 1), usually the lowest reflectance value) and the logistic equation had its starting point at 660 nm (Equation (Equation 3)).
(2)$$f\left(\lambda \right)=R_{550}+\lambda \;\cdot \;\beta$$
(3)$$f\left(\lambda \right)=\frac{R_{680}K}{R_{680}+\left(K-R_{680}\right)e^{-C\lambda }}$$

β slope coefficient;*R* Reflectance and $R_{\lambda }$ refers to the reflectance level at wavelength λ;λ wavelength (nm);*K* logistic function asymptote level or B_NIR_;*e* the natural logarithm base (or Euler’s number);*C* the logistic growth rate.

*Spectral Data Retrieval from Multispectral Camera*: For the data collection campaigns of 17 and 24 November 2018 (Figure 1-Elliot), the multispectral camera was employed to collect imagery at low-level flight (i.e., below 130 m altitude) within ±1 h of the solar noon and under clear sky conditions. Flight duration was always under six minutes due to the small area (0.1 ha) of the experimental area. The raw data was processed as per the Method “E” described in Poncet et al. [40]. The corresponding handheld sensor footprints were extracted from the imagery and their values were averaged. These averaged spectral responses were then employed as input in the hyperspectral simulation method, followed by a comparison with the handheld spectral data. For each date the number of observation pairs, between multispectral camera and handheld spectrometer, was equal to 180. Thus, in total, this analysis consisted of (*n*) 360 data points.

*Feature Engineering*: Features were generated as proposed in Kokaly [41] (i.e., continuum-removal) and Mutanga and Skidmore [16] (i.e., band-depth indices). The authors reported that, of these band-depth indices, the best performing of them were based on *normalized band depth index* (NBDI), which is calculated by subtracting the maximum band depth ($D_{c}$) from the band depth (BD) and dividing it by their sum. These indices can be generated through the use of hsdar::bdri for Band Depth Indices and hsdar::transformSpeclib for Band Area [42].

Additional features properties (hsdar::feature_properties) were generated as well as an optimized normalized ratio vegetation index (NRI; $\lambda {}_{1}$ = 745 nm and $\lambda {}_{2}$ = 755 nm), which has consistently performed well in related studies [31,32,33]. From the CR properties, both CR_area_^3^ and Max Depth Position^3^ (i.e., wavelength position of the maximum value observed in the feature) were employed (Table 1, Proposed Method).

Both CR_area_^3^ and Max Depth Position^3^ were employed through its cubic form to decrease the saturation effect. Finally, the logistic equation parameters found through the constrained optimization process (Equation (Equation 3)) were also included in the subset (Table 1, Feature Engineered).

The vegetation indices employed in the *current method* were Normalized Difference Vegetation Index (NDVI), Green Normalized Difference Vegetation Index (GNDVI), Normalized Difference Red Edge Index (NDRE), Normalized Green Red Difference Index (NGRDI), Leaf Chlorophyll Index (LCI), and Structure Intensive Pigment Index 2 (SIPI2) as well as the original band values (Table 1, Current Method).

Given the large amount of possible features generated by the *proposed method* and the collinear nature of simulated spectra, the derived indices were chosen based on the best performing indices reported in Mutanga and Skidmore [16] (continuum-removal based indices), and a Normalized Ratio Index optimized for pasture biomass estimation [31,32,33].

Besides the band values and coefficients found within the piecewise regression, additional vegetation indices were Band_area_, Optimized NRI_745–755_, Band_area_^3^, Max Depth Position^3^ (i.e., wavelength position of the maximum value observed in the feature), NDBI_744_, and NDBI_556_ (Table 1).

*Variable Filtering Selection*: For all features, including those developed in the *Feature Engineering* process, a minimal Pearson correlation coefficient filter of |0.2| was applied, following the methodology described in Alckmin et al. [48]. The remaining features were then employed in the modeling process (Table 1).

The formulas for the VIs can be found in the Appendix A. In Table 1, for the *current* methods, the references are next to each index, and for the *proposed* method references are listed in the “*Author (Reference)*” column.

#### 2.5.2. Biomass Modeling

Modeling (tidyverse) [49] was performed employing the workflow described in Kuhn and Wickham [50]. Random forest was chosen as the preferred regression algorithm for biomass retrieval as it was found to outperform other regression algorithms in a model performance study [31]. The data were split in a 75%/25% ratio (training/testing and validation, using location as a stratification factor—Figure 4-split). Explanatory values were centered and scaled. Hyperparameter tuning of the random forest algorithm was found through a grid-search approach of two hyperparameters (i.e., number of trees and number of randomly chosen at each split) in a *k*-fold cross-validation approach (Figure 4, Model Tuning). The chosen combination was within ten percent (tune::select_by_pct_loss) of the minimal Root Mean Square Error (RMSE) found within the search procedure.

The biomass modeling employed the same workflow, however, using two different subsets of VIs: (I) the use of original camera broadbands and VIs derived from these (referred to as *current method*) and (II) using the continuum-removed features and the parameters estimate through the two functions of the piecewise equation (referred to as *proposed method*). For further comparison purposes, the *proposed method* was reproduced using the simulated and original hyperspectral data.

#### 2.5.3. Performance Validation

**Model Performance**: Model performance was evaluated using two different strategies: (i) a cross-validation and (ii) and spatial cross-validation (Figure 4, Performance Assessment). These two strategies differ in regards to the composition of training and validation sets. In the first (Figure 4 (i), cross-validation), the algorithm was trained using the best combination of hyperparameters in the training dataset (75% of data). This model is then was assessed against a validation dataset equal to 25% of total observations (Figure 4, Model Tuning).

In the spatial validation, the original dataset was reduced so all locations have the same number of observations to prevent a class imbalance. Consequently, observations from Elliot were reduced from 1200 to 72. These observations were chosen at random while keeping the same biomass distribution as the original dataset. For the spatial validation, the algorithm was trained (using the same hyperparameter combination found in the model tuning) and assessed four times, always keeping a different location as the validation set (Figure 4-(ii) spatial validation).

**Feature Analysis**: Two unsupervised-learning visualization techniques were employed to provide insights about the dissimilarity among predictors and in relation to the predicted variable: the dendrogram and the heatmap, respectively. The clustering of the dendrogram was performed using the “Ward D2” hierarchical grouping technique [51]. The heatmap organized predictors so as to match the clusters provided by the dendrogram. Additionally, the observations (i.e., predictor values) were sorted in a descending order, based on its biomass weight. The color palette employed in the heatmap was scaled throughout predictors. The sorting (in descent order based on biomass values), alongside the scaled color palette, makes explicit the behavior and the relationship between different predictors. This analysis allows insights on how dissimilar are the predictors (i.e., dendrogram), while also showing how predictors behave throughout the biomass range and if such behavior differs among them.

## 3. Results

### 3.1. Reference Observations

**Biomass Measurements**: Although the number of samples ranges from 1200 (i.e., Elliot) to 72 (i.e., Goutum, Vreedepeel, and Zegveld), biomass ranges and distributions were comparable between all locations, with the exception of Vreedepeel. The Dunn test indicated that Vredepeel biomass distribution was not the same as other locations, as more than the 75% of samples present less 1.500 kg DM/ha. All locations had 75% (i.e., three quartiles) of observations below the 3000 kg · DM/ha (Figure 5a).

**Spectral Data**: The inspection of the principal component analysis (PCA) (Figure 5b) indicates that 98% of the MS variance could be decomposed in two principal components (i.e., Dim 1 and Dim 2). Observations were grouped per location (i.e., different colors and shapes) while different colored ellipses delimited 95% of observations of each location. These ellipsis were not separable (i.e., clusters), indicating that the grouping of spectra through its associated location has no distinguishable characteristics. In short, although each location has different attributes (e.g., soil type), by itself, the “location” factor has no influence in spectral-responses and cannot be used to point out differences in spectral response.

### 3.2. Data Analysis

#### 3.2.1. Hyperspectral Simulation

Hyperspectral information could be retrieved with a high degree of correlation and small overall error (Figure 6a,b). Overall, simulated and observed spectral information showed a degree of correlation above 99% (Figure 6). However, some areas (e.g., red-edge and near-infrared (NIR) plateau) consistently presented either over or underestimation of reflectance values, consequently indicating a higher level of RMSE. Yet, these average errors were usually below two percent (in reflectance values) (Figure 6a).

As the original reflectance values were retrieved with a high degree of accuracy, the CR could also be derived with satisfactory similarity to the original value (Figure 6b), allowing its use for the generation of predictors employed in the *proposed method*. As with reflectance, most of the simulated spectral range was highly correlated (i.e., above 97.5% correlation), although some areas (i.e., red-edge shift due to chlorophyll concentration) presented a higher level of error (i.e., RMSE).

Although minimal, an artifact was introduced at the breakpoint (λ = 685 nm) due to the discontinuity of values output by the two sub-functions. Such an artifact was more pronounced in the continuum-removed features, seen as a sharp transition in its maximum point (Figure 6b).

*Spectral Data Retrieval from Multispectral Camera*: For both dates (17 and 24 November 2018), there was a high level of agreement between the retrieved and observed spectral data, as presented in Figure 7. Overall, most bands presented a level of correlation above 80% and RMSE lower than 5% based on the (*n*) 180 observations per date. As suggested from Figure 6, the spectral regions corresponding to the red-edge inflection point (λ≈ 730 nm) presented the lowest level of correlation, whereas the NIR region presented the highest level of RMSE. In addition, the artifact at the breakpoint (λ = 685 nm) is noticeable. Between both dates, 17 November presented a better agreement between the retrieved and observed handheld spectral data (Figure 7-right side).

#### 3.2.2. Biomass Modeling and Performance Validation

*Feature Engineering and Filtering*: Within the *current* method, filtered VIs were SIPI2 and NGRDI. Furthermore, of the original MS bands, the B_green_ was excluded from the modeling dataset. Regarding the *proposed* method, the filtering process did not exclude any of the initial predictor set (Table 1).

*Hyperparameter Tuning*: The grid search of hyperparameters, both for the *current* and *proposed* methods, indicated that the less complex model (i.e., the one with a smaller number of trees and random variables per split) was within the ten percent RMSE model selection *tolerance* guideline (Figure 8). For all models, the derived hyperparameters were 500 for number of trees and 2 for randomly selected predictors at each node. Performances within the train–test sets, displayed a marginal improvement of the *proposed* method (Figure 8a,b,d,e in comparison to the *current* method (Figure 8c,f).

*Model Performance*: However, when applying the trained models to the validation set (Figure 9), in all cases, performance was equivalent. For the cross-validation approach, RMSE were close to 485–490 kg DM/ha. When employing the spatial cross-validation, performances decreased reaching values close to 610–620 kg DM/ha. Noticeably, in the spatial cross-validation (Figure 9d–f), there was a trend toward underestimation of biomass values for observations above 3500 kg DM/ha. This result, however, does not indicate a saturation of the prediction abilities of the model, as some observations in the 4000 kg DM/ha were accurately estimated.

#### 3.2.3. Features Analysis

Both Figure 10 and Figure 11 display the predictors clustered through a dendrogram (top-row), having at each end-leaf the corresponding correlation coefficient (linear relationship-R^2^). Within the dendrogram, the higher the node-split, the most dissimilar are the variables between each branch and leaves, showing that the most dissimilar predictors were *Band Area* and *Red*. A threshold (light gray-dashed line) was established, and each branch is colored according to this division.

Within the heatmap, observations were sorted from top to bottom (row-wise) in a descending fashion, according to its associated biomass weight (Figure 10 and Figure 11, bottom-right side). The value of each predictors was normalized (i.e., range equal to one) and colored accordingly (top-right side). On the right side of the figure, a scatterplot of the observations by corresponding biomass (kg DM/ha) is aligned (row-wise) with the corresponding predictors. As both sides of the figure are aligned, it is possible to observe the individual behavior of each observation in respect to the associated predictor values and biomass weight (Methods-Feature Analysis).

The dendrogram clustered the predictors in four main groups (different colors, according to the established threshold). The first and largest group (i.e., purple branch) is formed by Band Area (0.48), NDVI (0.43), Max Depth Position^3^ (0.44), C (0.29), Band Area^3^ (0.51), NDBI_556_ (0.50) and GNDVI (0.48)—predictors and determination coefficients, respectively. In essence, this group clustered the predictors which were related to the VIS and NIR features and those related with CR features, and commonly associated with plant structural properties.

The second group (i.e., light-green branch) corresponds to the predictors with the highest correlation with biomass, including the Optimized NRI_745–755_ (0.55), NDRE (0.53), Leaf Chlorophyll Index (LCI) (0.55), NDBI_744_, which are commonly associated with plant greenness and different nitrogen concentrations. The following group (i.e., dark-green branch) is made of two of the original bands Red-Edge (0.35) and near-infrared (NIR) (0.48). The final group (i.e., pink branch) presents the lowest correlation with biomass, and corresponds to the Beta (0.38) and Red (0.23). All four main branches present predictors from both *current* and *proposed* methods.

The proximity between leaves indicates a similar response-pattern between predictors and, consequently, biomass. Accordingly, distant branches correspond to less similar clusters. The branches containing the original band values (i.e., dark-green and pink) are distant from VIs, displaying the two highest nodes.

Furthermore, variables within the same branch display the same color patterns, indicating its high level of collinearity. Finally, the response (color) pattern for the data collected in the Netherlands (Figure 11) is equivalent to the Australian data. In none of these heatmaps is it possible to distinguish a particular behavior linked to Location. Through the visual inspection of the color pattern, it is possible to identify saturation effects (e.g., NDVI, GNDVI, and B_red_) of the predictors above 1500 kg DM/ha. In a opposite way, the indices clustered under the light-green branch (i.e., Optimized NRI_745−755_, NDRE, LCI and NDBI_744_) display no meaningful sign of saturation, reaching its maximum levels (i.e., darker-blue) at biomass levels above 3500 kg DM/ha.

## 4. Discussion

Our results have shown that a satisfactory reconstruction of the hyperspectral signature of a vegetated canopy can be retrieved from multispectral measurements compatible with a commercially available sensor (Figure 6a). Such is to be expected, as the overall shape of the function (i.e., spectral response) is known and adequate functions were employed in the appropriate spectral ranges. Consequently, this study has also shown that a straightforward parametric equation can be employed for such purposes, without resorting to prior information as necessary for Bayesian or multivariate approaches (Results-Hyperspectral Simulation). When tested against a real multispectral camera, although with a smaller sample size (two different dates, *n* = 360), the hyperspectral retrieval method proved to be effective, reaching acceptable correlation and RMSE levels (above 80% and below 5%, respectively; Figure 7).

Furthermore, this study has shown that synthetic spectral features, such as continuum-removed features, can also be retrieved with a high degree of accuracy (Figure 6b). Moreover, models that employed either observed or simulated spectral data have performed equally well, achieving a RMSE of around 490 kg DM/ha, similar to values reported in Thomson et al. [52] and Alckmin et al. [31]. The accuracy achieved reflects a long-term study, lasting close to two years, two different countries and with 1416 spectra-sample pairs.

Nevertheless, the results have shown that more elaborate VIs, such as those in the *proposed method* and originally introduced in Mutanga and Skidmore [32] and Mutanga and Skidmore [16], perform equally as well as those in the *current method*. Thus, it seems fair to state that “*while there is a long tradition of creating vegetation indices, there is also a tradition of proving these functionally equivalent*” [53]. Such a statement is corroborated through the analysis of dendrograms and heatmaps, where different VIs from both methods are clustered side-by-side within the same branches.

Both a narrowband optimal index (i.e., Optimized NRI) and a broadband index (i.e., NDRE), displayed almost identical correlation levels with biomass (Figure 10 and Figure 11). Furthermore, both dendrogram and model tuning indicate that the number of predictors and model complexity could be decreased as no significant improvement was found when using more complex parameters (Figure 8) and many indices were shown to be redundant (Figure 10 and Figure 11).

Overall, results indicate that the maximum accuracy can be found using straightforward regression techniques, with a low risk of incurring in underfitting issues due to conservative hyperparameters employed (Figure 8). Consequently, it is reasonable to point that future research may explore less complex versions of the algorithm employed in this study: random forests. A number of different tree-based algorithms, may take advantage of nonlinearity provided by this family of algorithms, while using a small number of predictors and less complex model structures. Such would may improve the overall understanding of the underlying phenomena, enhance its explainability to a broader audience and provide a higher scalability of methods to larger datasets while decreasing the necessary processing-power to deploy these models.

The range of error metrics found in the validation set are in line with previous research [31,52]. Based on the method and experimental design, an RMSE error equivalent to 460–495 kg DM/ha is equal to 7–7.5 g within the footprint. Although strict control was employed, such a small value may partially be due to different error sources, such as weighing scale precision, small imprecisions in cutting height, or incomplete harvesting. Such a systematic error should be normally distributed throughout all levels of biomass. However, larger residuals are found in high biomass levels (i.e., above 3000 kg DM/ha, indicating that other confounding factors (e.g., difference in scene support or ambiguous spectral response) may be decreasing achievable accuracies. A possible explanation lies in the well-known asymptotic nature of reflectance [54] by which a denser or thicker optical medium has decreasing reflectance response and, thus, further changes in the target do not equate to changes in reflectance levels (i.e., saturation). This heteroscedastic trend would be even more explicit if a larger percentile of biomass samples were above this saturation threshold. However, under operational scenario grazing or harvest of pasture are usually performed before this biomass range is reached.

From a biological perspective, two main broadband spectral features are acting within the spectral range (i.e., 550–790 nm) employed in this study: chlorophyll content and leaf area index. While chlorophyll concentration is related positively to a minor deepening and a major widening of the absorption features, higher levels of LAI provide an asymptotic increase in reflectance level at the NIR plateau. These two phenomena explain the success in the retrieval of the hyperspectral data through a piecewise function and indicate that the ability to predict pasture biomass is linked with the ability to predict both phenomena regardless of the subset of vegetation indices employed to such end.

From a sensors design perspective, this study also suggested that the use of hyperspectral sensors in outdoor environment is of limited utility when not employed for measurements of narrowband absorption features. As an example of such, the simulated spectra cannot account for the fluorescence peak at around 760 nm or the red-edge position [55] at 700–725 nm (Figure 6a). Other areas of the spectra could be simulated with a high level of accuracy, thus not requiring a dedicated sensor.

As limitations of this study, although the handheld instruments employed are considered the benchmark for spectral measurements, commercial multispectral cameras can introduce instrumental error, due to poor design and radiometric processing, which may hamper the accurate retrieval of the spectral information.

Despite being outside the scope of this study, the difference in between the simulated and camera-retrieved spectral data can be explained through differences in sensor-viewing geometry, illumination angle, and radiometric processing of the multispectral data, as well as differences in spatial support (i.e., differences related to the point-spread-function of the handheld instruments or regarding the footprint averaging of the multispectral imagery-Mac Arthur et al. [56]). Additionally, the radiometric consistency of the multispectral camera has been reported as sub-optimal [57], possibly contributing to a fraction of the error. There was also a noticeable difference between dates, suggesting the need for further research on protocols for data collection of multispectral imagery at low-level flight.

As additional limitation, soil spectral information has not been considered. This, however, did not impact the accurate retrieval of the original hyperspectral data. For most operational scenarios, grazing moments and pre-grazing biomass targets are reached after 95% light interception [58]; thus, target background, such as soils, has a negligible influence in the final spectral signature. The method proposed for spectral retrieval would be of limited utility for low-levels of biomass (e.g., below 500 kg DM/ha).

Conversely, it seems reasonable to hypothesize that, as reported in Tucker [54], at high levels of LAI, irradiance has little to no interaction with the lower strata of the canopy. Thus, for high levels of biomass, empirical relations between spectra and canopy properties have found its accuracy ceiling. Complex algorithms cannot improve performance due to an absent relationship between predictors and output. For this circumstances, accuracy improvements should come different sources of data or from different modeling strategies, such as time-series analysis.

Finally, this study successfully compared different validation strategies, both through a hold-out validation set and spatial validation. These strategies differed in terms of accuracy, indicating that some variability is introduced in the relationship of spectra biomass. Yet, there is no strong indication that location should be considered as a necessary predictor to improve accuracy, given the PCA analysis results. Furthermore, while there is a difference of around 120 kg DM/ha when employing spatial validation, the same rational also shows that there are larger (i.e., 490 kg DM/ha, validation) sources of errors not specifically related to location.

## 5. Conclusions

This study has shown that hyperspectral data of vegetated canopies, in specific of perennial ryegrass, can be retrieved from multispectral measurements through the use of a straightforward piecewise parametric equation. As a limitation, however, discrete narrowband absorption features, such as fluorescence, could not be accurately retrieved. Vegetation indices and other spectral features derived from the simulated spectra have not improved model performance for biomass estimation when compared to indices already employed in current practice. This may be due to the collinear nature of spectra and derived indices or transformations, introduced by the parametric equations.

The maximum achievable accuracy (i.e., RMSE) was equal to approximately 490 kg DM/ha. This error level may be used as a benchmark for spectra-based models when employing top-of-the-canopy reflectance (λ = 550–790 nm). Such a limitation may be due to the asymptotic response-nature of reflectance and high levels of biomass (LAI > 4), and to uncontrolled sources of error (such as scene support) not accounted for in this study.

Finally, the results indicate that, in outdoor conditions, a multispectral sensor can perform equally as well as a hyperspectral sensor when aiming to retrieve pasture biomass estimations and the spectral range employed is limited to 550–790 nm. In this range, the main reflectance drivers are chlorophyll absorption and cell wall reflection, both broadband phenomena and measurable through a multispectral sensor.

Future research should focus on additional uncorrelated predictors not correlated with spectral response, such as weather data; different modeling strategies, such as time-series or forecast models; and testing whether less-complex machine learning algorithm could provide the same level of error, while decreasing processing power requirements for model deployment.

## Figures and Tables

**Figure 1 sensors-20-07192-f001:**
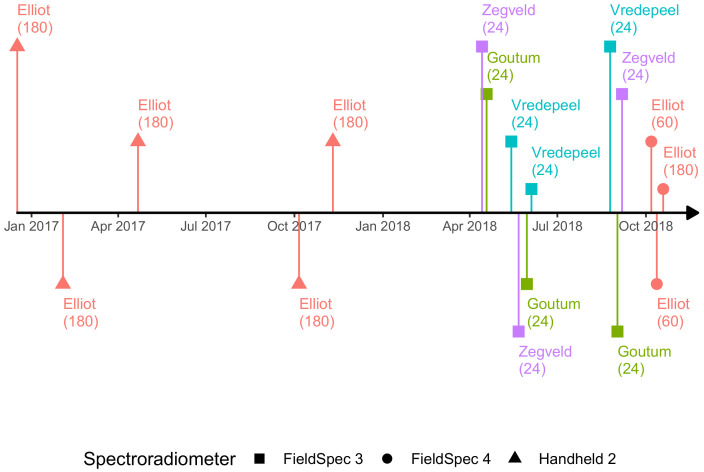
Timeline of data collection campaigns. Each campaign is coded by location (color), number of observations (within brackets), and instrument (spectroradiometer) employed (shape). First date collection campaign took place in 18 December 2016 and the last in 24 November 2018.

**Figure 2 sensors-20-07192-f002:**
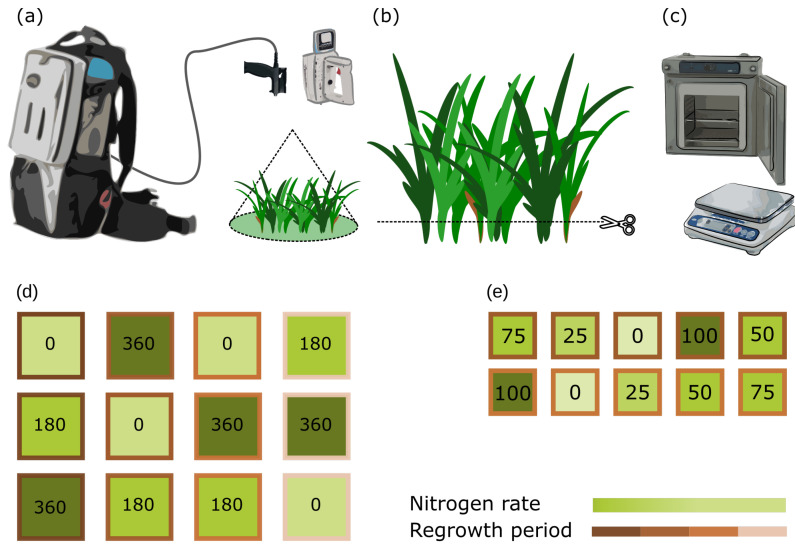
Data collection protocol (**a**–**c**) and plot layout (basic pseudo-replicate, (**d**,**e**)). Top: (**a**) Spectral Measurements, (**b**) Mechanical Defoliation and (**c**) Drying and Weighing. Bottom: basic Dutch (**d**) and Australian (**e**) plot layout. Borderline colors indicate regrowth period and hues of green (darker = higher rates) indicate nitrogen levels (rates are also indicated within each plot).

**Figure 3 sensors-20-07192-f003:**
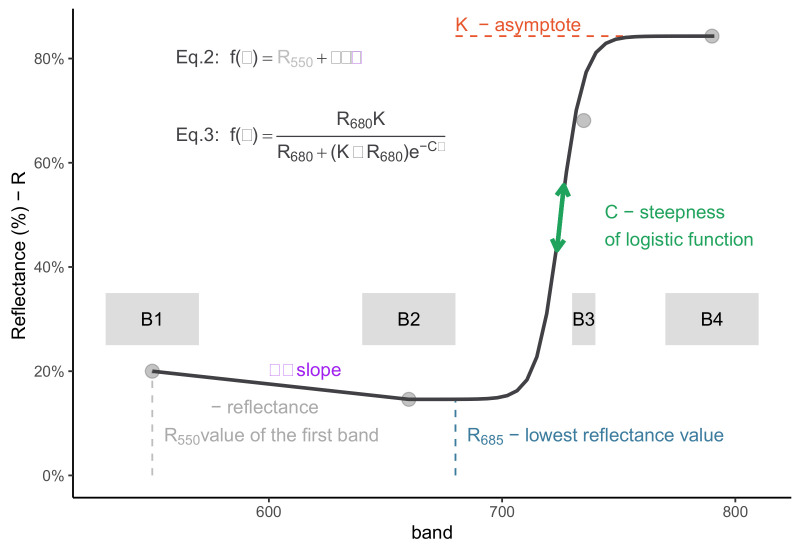
Hyperspectral simulation method. A piecewise equation Equation (Equation 1), using two different function: linear function Equation (Equation 2) and a logistic function Equation (Equation 3). Parameters were found through a constrained optimization procedure. Multispectral bands (B_1−4_), corresponding to a common UAV multispectral sensor) displayed in grey shaded (width) area and grey circles (center).

**Figure 4 sensors-20-07192-f004:**
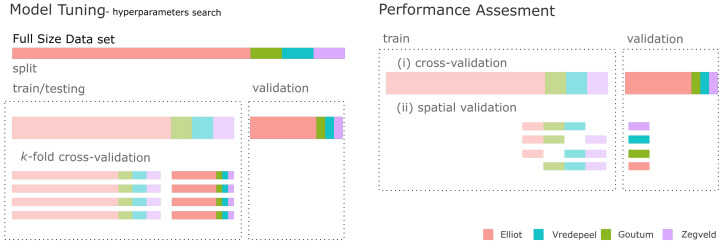
*Biomass Modeling* (**left**) and *Performance Validation* (**right**). Modeling: the original dataset was split in a train/test dataset (75%) and a validation dataset (25%). Tuning of hyperparameters was performed through a *k*-fold cross-validation approach. The best tuning (i.e., workflow) was then used in a *Performance Assessment* and validated through a (i) validation set or (ii) spatial validation strategy.

**Figure 5 sensors-20-07192-f005:**
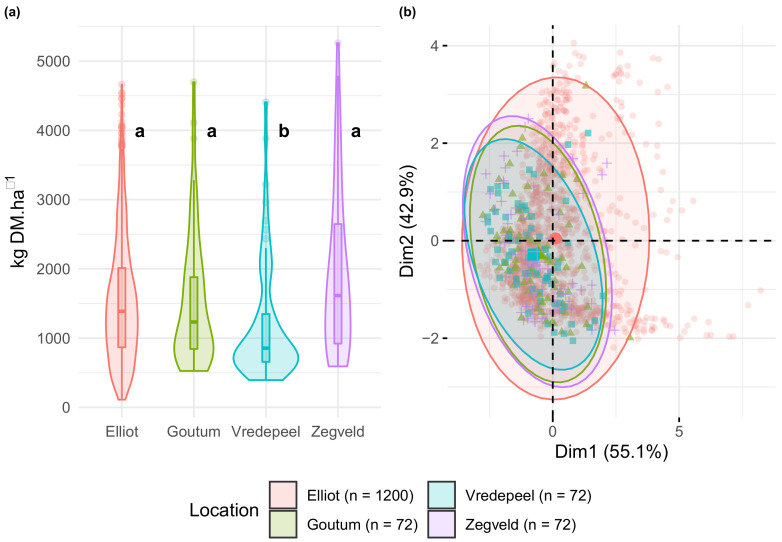
(**a**) Boxplot and violin plot of observed sample biomass grouped by location. Dunn’s Multiple Comparisons (α-level = 0.05) results presented next to each boxplot. Number of observations (*n*) per location within the bottom legend. (**b**) PCA of multispectral sample measurements. Ellipsis represent the centers of mass of each location and cover 95% of observations per group.

**Figure 6 sensors-20-07192-f006:**
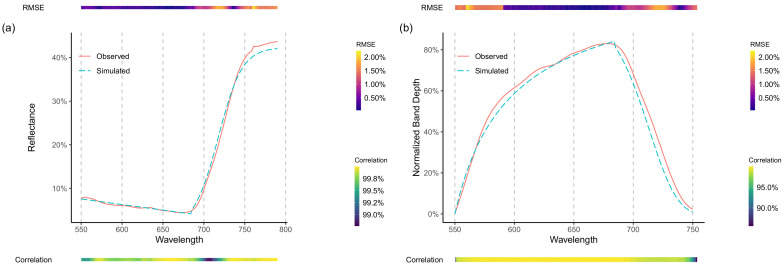
Observed and Simulated Reflectance (**a**) and Band Depth comparison (**b**). In each figure the corresponding average of all *Observed* and *Simulated* spectra are depicted. A top and bottom bar present both the RMSE (top) and correlation (bottom), per wavelength, between the *Observed* and *Simulated* spectra.

**Figure 7 sensors-20-07192-f007:**
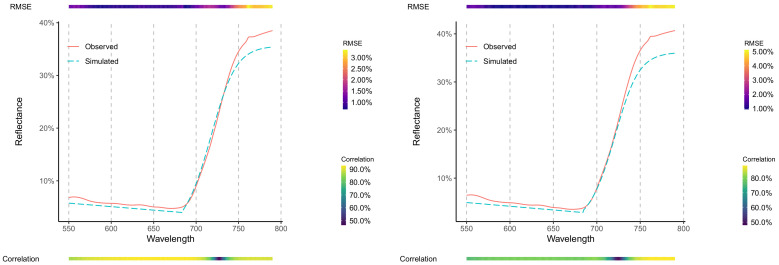
Observed Hyperspectral (solid red line) and Hyperspectral Retrieved (dashed blue line) from Multispectral Camera for the 17 (**left**) and 24 of November 2018 (**right**). In each figure, the corresponding average of all *Observed* and *Retrieved* spectras are depicted (*n* = 180). A top and bottom bar present both the RMSE (top) and correlation (bottom), per wavelength, between the *Observed* and *Retrieved* spectra.

**Figure 8 sensors-20-07192-f008:**
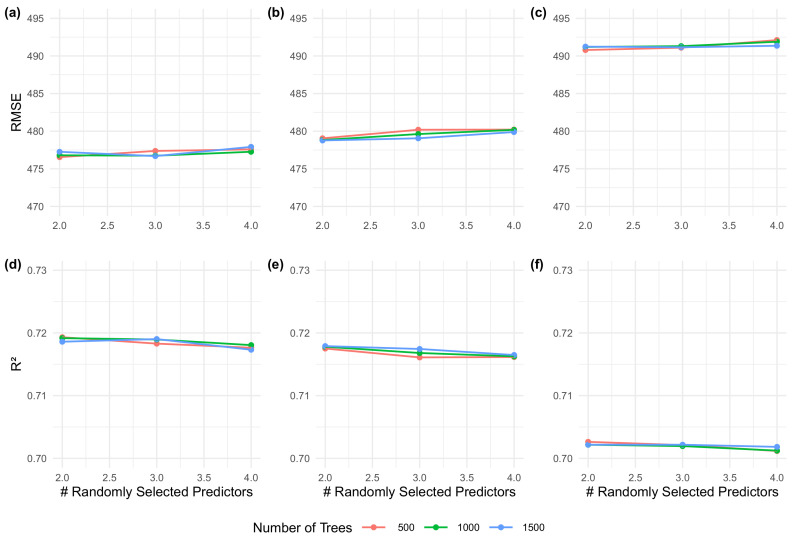
Grid-search hyperparameter tuning. Different coloured lines represent different number of trees and, the bottom-axis the number of randomly chosen predictor at each node. Sub-figures (**a**,**d**) are associated with training-testing values found for the *Proposed* method based on observed hyperspectral data, while sub-figure (**b**,**e**) correspond to the simulated hyperspectral data. Sub-figures (**c**,**f**) represent training-testing results for the *Current* method.

**Figure 9 sensors-20-07192-f009:**
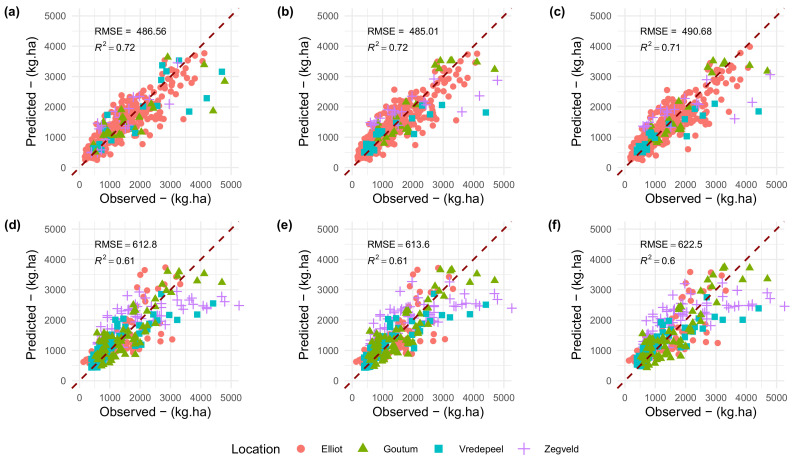
Predicted by Observed scatterplots. Validation strategy presented in a row-wise fashion. Accordingly, columns corresponds to the Methods employed. Top row corresponds to cross-validation results. Bottom row corresponds to spatial validation strategy. Methods: (**a**–**d**) Hyperspectral data, (**b**–**e**) *Proposed Method*, and (**c**–**f**) *Current method*.

**Figure 10 sensors-20-07192-f010:**
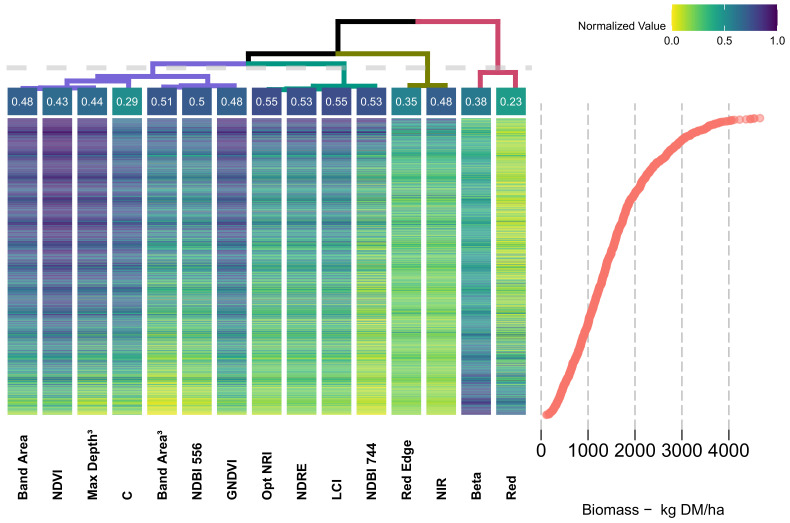
Dendrogram and clustering of features: Australia. Top left-side: dendrogram clustering of predictors, each end-leaf express the R^2^ relationship with Biomass. Top right-side: color palette legend for the heatmap. Bottom left-side: heatmap of predictor variables, organized column-wise as per Ward-D2 criteria and row-wise in a descending order of biomass weights. Bottom right-side: scatterplot of the observations (i.e., samples) biomass weight, arranged in descending order.

**Figure 11 sensors-20-07192-f011:**
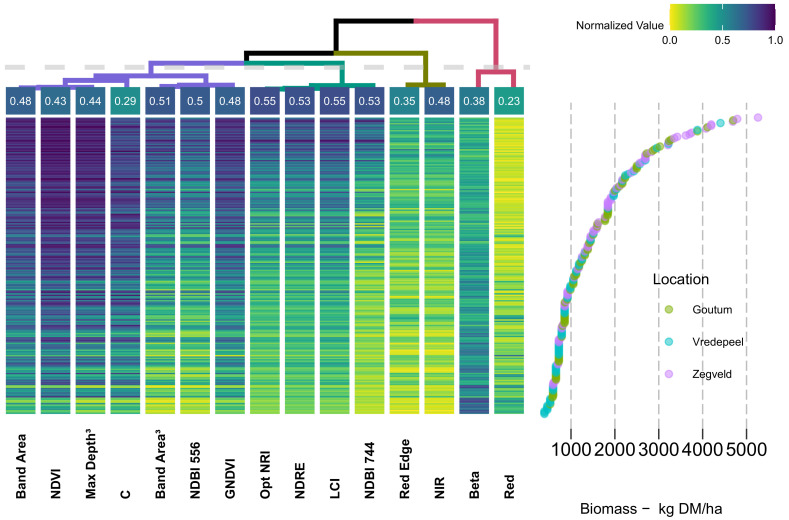
Dendrogram and clustering of features-Netherlands. Top left-side: dendrogram clustering of predictors, each end-leaf express the R^2^ relationship with Biomass. Top right-side: color palette legend for the heatmap. Bottom left-side: heatmap of predictor variables, organized column-wise as per Ward-D2 criteria and row-wise in a descending order of biomass weights. Bottom right-side: scatterplot of the observations (i.e., samples) biomass weight, arranged in descending order.

**Table 1 sensors-20-07192-t001:** Table of predictors divided per method.

Current Method	Proposed Method	Author (Reference)	Original Bands
GNDVI [43]	Optimized NRI _745–755_	Mutanga and Skidmore [32]	B_Green_ *
LCI [44]	NDBI_556 or 744_	Mutanga and Skidmore [16]	B_Near Infrared_
NDRE [45]	Max Depth Position^3^	Feature Engineered	B_Red_
NDRGI * [46]	C and Beta	Feature Engineered	B_Red-Edge_
NDVI [7]	Band_area_^3^	Feature Engineered	
SIPI2 * [47]	Band_area_	Mutanga and Skidmore [16]	

* Predictors marked with an asterisk were eliminated in the Filtering Process.

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
