# Peer review of "Retrieval of Hyperspectral Information from Multispectral Data for Perennial Ryegrass Biomass Estimation"

_sensors, 2020, doi:10.3390/s20247192_

Round 1
Reviewer 1 Report
Dear Authors
The manuscript presented a systematic methodology to synthesis the hyperspectral reflectance data from multispectral data. It is a good initiative to model hyperspectral data because currently there is high potential to acquire UAV based multispectral data which can be ultimately synthesis to hyperspectral data for deep analysis. I found this manuscript is touching a novel area in remote sensing.
However, I have some concern regarding the methodology and interpretation of the results.
- The multispectral data was generated by spectral resampling the original field spectroradiometer data. However, it would be great to see at least in an experimental plot level, how this resampling is accurate with the actual measurement from the multispectral camera data. Because later authors use resampled multispectral data as an input synthesis hyperspectral data without concerning the spectral resampling error, as well as other measurement errors.
- Did authors assume that there is no effect from the soil reflectance to the measured signal? If no, how it is explaining when you synthesis the hyperspectral data from two equations.
- Why the authors specifically use random forest regression out many machine learning regression algorithms?
- It is not clear how exactly the variable filtering was done. So please elaborate it little further.
- Table 1 is not clearly understood. Does each row contain records of the same data or each column has some discrete info? Please add equation for the VIs at least in Appendix.
- It is not clear how many variables after filtering that used in each modelling method (hyperspectral, current, proposed). Please add a table with used variables after filtering for model building.
- According to Figure 8, there are no substantial differences in prediction accuracies among the methods, is it due to a) bias from original hyperspectral data which later used to resample multispectral data and the synthesis hyperspectral data or b) random effect?
- Also, the results indicated that multispectral data without hyperspectral features can have a similar prediction accuracy of biomass. How it was resulted in the previous studies? That part is missing in the discussion.
- Figure 7, please use the common y scale, which helps easy comparison. Further, it seems that the default hyperparameter (10.1002/widm.1301) values were okay with this data set. So this part can be clearly avoided to simplify the complexity of the manuscript.
- It is difficult to understand the concept of Fig 9 & 10. Specifically the graphs in the right. Is it cumulative biomass?
- As authors claimed that there is no difference between Australia and Netherlands data, why then the authors have drawn separately Dandegram to explain variables for two countries?
- Since this manuscript focused on biomass estimation of perennial ryegrass, I think the title should indicate that instead of general pasture biomass.
- Why authors did not analyse the model prediction capability against the site as well as the fertiliser treatment?
Good Luck
Author Response
Dear Reviewer,
We would like to thank you for the careful revision. Please find in our rebuttal, where we have considered all the points raised and incorporate, when possible and appropriate, all of them.
Kind regards,

Reviewer 2 Report
The manuscript entitled ‘Retrieval of hyperspectral information from multiespectral data for pasture biomass estimation’ is an interesting article dealing with the necessity of fast and accurate estimations of fields in order to improve pasture management. The data base used is very representative, using data from 4 different locations from two different countries and at three different seasons, improving its quality and representativeness. The manuscript is well written and deserves to be published after minor changes.
Some minor suggestions can be found here:
L16: Keywords should provide additional information to the title for searching porpoises, so please, consider avoiding repeated words as ‘hyperspectral’ or ‘biomass’.
Figure 8: The model seems to tend to saturate at 2500 kg/ha but it seems that this is not well represented in your regressions as most of the data observed were smaller than 2000 kg/ha. Perhaps this limitation should be discussed deeper in the manuscript.
L310-313: these indices are frequently related to structural properties of the plant, and mostly related to LAI. Probably you should discuss deeper based on other authors information.
L315-318: these other indices are frequently related to nutritional status of the plant (greenness and N content). Probably you should discuss deeper based on other authors information.
Author Response

(The authors gave the same response as above.)

Round 2
Reviewer 1 Report
Dear Authors
All the reasons explained by the authors acceptable, as well as the changes done to the manuscript.
Author Response
Thank you for your review of our manuscript and your valuable comments on our paper.